# The Prevalence of Childhood Asthma, Respiratory Symptoms and Associated Air Pollution Sources Among Adolescent Learners in Selected Schools in Vhembe District, South Africa

**DOI:** 10.3390/ijerph21111536

**Published:** 2024-11-20

**Authors:** Funzani Rathogwa-Takalani, Thabelo Rodney Mudau, Sean Patrick, Joyce Shirinde, Kuku Voyi

**Affiliations:** 1Department of Advanced Nursing Science, Faculty of Health Sciences, University of Venda, Thohoyandou 0950, South Africa; 2School of Health Systems and Public Health, University of Pretoria, Pretoria 0001, South Africajoyce.shirinde@up.ac.za (J.S.);; 3Centre for Environmental and Occupational Health Research, School of Public Health, University of Cape Town, Cape Town 7925, South Africa; thabelo.mudau@uct.ac.za; 4University of Pretoria Institute for Sustainable Malaria Control, University of Pretoria, Pretoria 0001, South Africa

**Keywords:** asthma, adolescents, indoor residual spraying, respiratory symptoms, ISAAC

## Abstract

This study investigated the prevalence of childhood asthma and respiratory symptoms with their associated air pollution sources among adolescents aged 13–14 years residing in a Malaria-endemic region. Methods: A cross-sectional survey was conducted with 2855 adolescents from fourteen (14) selected schools in communities exposed to high levels of air pollution from indoor residual spraying (IRS) that is used for malaria vector control in the Vhembe region. Data were collected using a self-administered standardized International Study of Asthma and Allergies in Childhood (ISAAC) questionnaire. Statistical software STATA version 17 was used to analyze the data. Binary logistic regression was used to determine the relationship between air pollution sources and childhood asthma/symptoms. Results: The prevalences of asthma, ‘wheeze ever’ and ‘wheeze in the past’ were 18.91%, 37.69% and 24.69%, respectively. The results from the adjusted binary logistic regression model indicated that exposure to tobacco smoke (OR = 1.84; 95% CI: 1.08–3.16), smoking a water pipe (OR = 1.65; 95% CI: 1.16–2.36) and the use of paraffin as fuel for heating (OR = 1.70; 95% CI: 0.97–2.88) and cooking (OR = 0.48; 95% CI: 0.29–1.00) were significant risk factors for asthma. Trucks passing through the streets, having a cat at home and using open fires were significantly associated with ‘wheeze in the past’. Finally, using gas for cooking (OR = 0.72; 95% CI: 0.53–0.99), open fires for heating (OR = 0.53; 95% CI: 0.35–0.80) and smoking a water pipe (OR = 2.47; 95% CI: 1.78–3.44) were associated with ‘wheeze ever’. Conclusions: School children living in these communities had an increased risk of developing asthma and presenting with wheezing due to exposure to environmental air pollution sources.

## 1. Introduction

Air pollution is the second most significant health risk in Africa, contributing to 1.1 million deaths in 2019, and children with asthma are particularly vulnerable to its effects [1]. Many studies have shown a significant link between exposure to air pollution and the risk of developing respiratory diseases or illnesses, and these include asthma and wheezing [2,3,4,5].

Worldwide, asthma alone impacts 300 million people and over 250 million people experience wheezing [6,7,8]. Asthma is one of the most common and significant non-communicable respiratory diseases, characterized by the swelling and narrowing of airways, which makes breathing difficult [9,10,11]. Wheezing in children is a common condition characterized by a continuous high-pitched sound that emanates from the chest during exhalation, and is a hallmark symptom of asthma [12]. Furthermore, wheezing is often one of the first symptoms of airway obstruction, which indicates the need for further investigation. This relationship is significant as the early recognition and management of wheeze may help prevent the advancement of asthma in children.

Global asthma prevalence ranges from 9.1% to 9.5% for children and from 9.1% to just over 10 percent in adolescents [13,14]. The global burden of disease report highlighted an increase in asthma and respiratory symptom prevalence in children [10]. Furthermore, a study conducted in Iran also showed an increase in asthma symptoms especially in children and adolescents [13,15]. Numerous epidemiological studies have reported increased prevalence rates of asthma and wheezing symptoms in adolescents residing in regions with increased air pollution levels [5,7,10,16].

The onset of asthma and wheezing is likely attributable to a complex interaction of environmental and lifestyle factors, rather than solely individual characteristics, highlighting the necessity of investigating the relationship between air pollution and asthma in extensive, representative populations [7,11,16]. Primary constituents of air pollution in homes originate from combustion sources, including wood, gas, coal and paraffin stoves utilized for cooking, heating and illumination [17]. In many communities like Vhembe, in middle-income countries, many people still rely on solid fuels to heat their homes.

Another study also reported that being male, having pets at home, dietary patterns, and exposure to environmental factors are associated with an increased prevalence of asthma [18,19]. Furthermore, an epidemiological study showed a strong association in the multivariate analysis between present asthma and smoking, female gender, pet exposure and elevated socioeconomic level [20]. Additionally, severe asthma correlated with smoking, exposure to pets, outdoor pollutants and informal housing [20]. The effects of Exposure to Environmental Tobacco Smoke (ETS) on health have also been excessively reported in the literature, with some adverse effects implicated in birth defects, neurological defects, impaired lung function, asthma and wheezing [17,21,22].

The prevalence of asthma respiratory symptoms in Vhembe District is expected to currently be higher, especially in selected communities, due to exposure to indoor residual spraying (IRS), which is used for Malaria control. IRS is a key malaria control strategy involving the spraying of pesticides on indoor surfaces to kill mosquitoes, especially in malaria-endemic regions such as Vhembe District in South Africa. Although IRS has been proven beneficial in reducing the transmission of malaria, inhaling the complex chemical particles used may pose a risk to vulnerable populations, especially children and the elderly.

Moreover, other air pollution sources such as dusty gravel roads, the use of fire for cooking or heating, tobacco smoking and other outdoor sources may be contributors to air pollution [23]. Despite the presence of overwhelming evidence that air pollution affects human health, some significant challenges remain, and one of the biggest challenges in resolving the human health and air quality relationship is the collection of reliable health data from a broad sample of the population.

ISAAC studies conducted in South Africa have addressed the prevalence of asthma, wheeze and associated risk factors in mining and industrial areas. However, to the best of our knowledge no study has assessed the prevalence of asthma and respiratory symptoms and associated risk factors in Vhembe district, an area exposed to high pollution levels. The aim of this study was to investigate the prevalence of asthma, respiratory symptoms and their associations with air pollution sources.

## 2. Materials and Methods

### 2.1. Procedure

A cross-sectional study among 13- to 14-year-old school children was conducted at 15 randomly selected schools located in Vhembe district in Limpopo Province. Seven schools were primary schools, and eight schools were secondary schools.

Prior to the study commencing, school principals were contacted and the study’s aim and objectives were critically discussed. School principals gave informed consent agreeing that their school would participate in the study. Upon consenting, a suitable date for data collection at the school was set. All eligible school children were given informed consent forms for parents, guardians or caregivers to complete, and along with that document was an assent form for the children themselves to complete.

Permission to conduct the study was obtained from the Limpopo Provincial Department of Education and the Vhembe District Department of Education, where the schools were located. The Research Ethics clearance for this study was granted by the University of Pretoria Ethics Committee and the Limpopo Provincial Research Ethics Committee under registrations (REC 482/2022) and (LPREC/54/2022:PG), respectively.

### 2.2. Inclusion and Exclusion Criteria

School principals and school children that showed lack of interest or did not give consent were excluded from the study. Furthermore, if parents, guardians or caregivers did not give consent for the children to participate in the study, this resulted in those children being excluded from the study. Figure 1 below shows a flow chart of the procedure followed by the researcher to recruit participants to this study and the participation rate thereof.

### 2.3. Questionnaire

Data were collected using the English version of the previously validate ISAAC written questionnaire, and school children completed the survey during school hours in the Life Orientations period. The medium of instruction used during the data collection at the schools was English.

### 2.4. Health Outcomes

The health outcome ‘asthma ever’ was estimated by a positive response to the question “Have you ever had asthma?”

The health outcome ‘wheeze ever’ was estimated by a positive response to the question “Have you ever had wheezing or whistling in the chest at any time in the past?”

The health outcome ‘wheeze in the past’ was estimated by a positive response to the question “Have you had wheezing or whistling in the chest in the past 12 months?”

### 2.5. Confounders

The following variables were identified as potential confounders: gender (female/male), community (exposed/non-exposed), born in the study area (yes/no), school (primary/secondary), being a twin (yes/no), type of fuel used in the house for cooking (electricity/gas/paraffin/open fire—wood/coal), type of fuel used in the house for heating (electricity/gas/paraffin/open fire—wood/coal), having pets (dog and/or cat) at home (yes/no), the frequency of trucks passing in the street (never/seldom/frequently during the day/almost whole day), currently smoking tobacco (not at all/less than daily/daily), smoking a water pipe (yes/no), vigorous physical activity (never or only occasionally/once or twice per week/three or more times a week), use of paracetamol (never/at least once a year/at least once a month), playing social games including social media and/or watching television (less than 1 h/1 h but less than 3 h/3 h but less than 5 h/5 h or more).

### 2.6. Data Management and Statistical Analysis

The data were captured using Research Electronic Data Capture (REDCap), where a template of the questionnaire was created and responses were captured by qualified data capturers to ensure the quality of data. Following data capturing, the data were analyzed using statistical software STATA Version 17. The main purpose of descriptive statistics in this study was to summarize and organize the data. The descriptive statistics employed in this study were measures of frequency, whereby we measured how often a response was given in each variable in the form of count, percentage and frequency. This was performed for categorical data. The prevalence of asthma and respiratory symptoms/illnesses and proportion of pollution sources and confounding variables were calculated by dividing the number of participants who responded “YES” to a particular question by the total number of questionnaires completed. Where observations were not recorded, they were deemed as missing data, which consequently caused a difference in the sample size.

In this study, we assessed the association between air pollution sources and three health outcome variables: ‘asthma ever’, ‘wheeze ever’ and ‘wheeze in the past’. To estimate the associations between air pollution sources and the health outcome variables, univariate and binary logistic regression analysis was used. Univariate logistic regression was initially performed to assess the individual effects of each predictor variable on the outcome, and that was presented by unadjusted odds ratios. Subsequently, binary logistic regression was used to control for potential confounders = 8 and to determine the adjusted odds ratios for the relationships between risk factors and outcome variables. We specifically employed binary logistic regression due to the dichotomous nature of the outcome variables (‘asthma ever’: yes/no; ‘wheeze ever’: yes/no; ‘wheeze in the past’: yes/no). Crude and adjusted odds ratios, along with their 95% confidence intervals, determined the strength of the association.

## 3. Results

Table 1, below, shows the total response rate of 86.5%, with the majority of study participants being female (57.89%). A proportion of nearly 70% of the study participants were 13 years old and 61.46% lived in communities exposed to indoor residual spraying (IRS) for malaria control. Seventy-six percent of study participants were born in the area and nearly 55% were in secondary school.

### 3.1. Health Outcomes and Risk Factors of the Study Participants

The prevalences of the three health outcome variables (‘asthma ever’, ‘wheeze ever’ and ‘wheeze in the past’) were 18.91%, 38.06% and 67.40%, respectively, as shown in Table 2 above. Table 3 shows that the main residential fuel used for cooking and heating was electricity. However, a notable number of study participants still used air pollution sources such as gas (14.04%), paraffin (4.04%) and open fires (23.73%) for cooking. Air pollution sources frequently used for heating were gas, with nearly 12%, paraffin (14.78%) and open fires (6.83%). Furthermore, majority of the study participants were currently not smoking tobacco at all, while roughly 5% said they smoked daily. Nearly 11% of the study participants affirmed that they smoked a water pipe, as shown in Table 3.

### 3.2. Univariate Regression Analysis

Several factors were found to be significant for ‘asthma ever’ in the univariate regression analysis in Table 4, as follows: engaging in physical activity once or twice a week (OR 3.83 (3.08–4.77)) or three or more times a week (OR 3.15 (2.33–4.26)), watching TV for more than one hour but less than three hours (OR 1.35 (1.08–1.68)), spending three hours or more on social games (OR 1.75 (1.35–2.27)), being a twin (OR 3.62 (2.78–4.70)), trucks seldom passing through on weekdays (OR 1.46 (1.12–1.90)), frequently throughout the day (OR 2.52 (1.94–3.26)) and almost the whole day (OR 1.65 (1.25–2.16)).

Using paracetamol (OR 2.50 (1.90–3.29)), having a dog or cat at home (OR 2.22 (0.81–2.71), OR 1.44 (1.18–1.75)), exposure to tobacco smoking or products ((OR 2.28 (1.54–3.26), OR 2.62 (2.01–3.42)), the type of fuel used for cooking (paraffin OR 1.93 (1.49–2.51) and gas OR 3.90 (2.61–5.82)) and the fuel type used for heating (gas OR 2.09 (1.60–2.72)) were significant factors for developing asthma.

In the adjusted model, also in Table 3, engaging in vigorous physical activity once or twice a week/three or more times a week was significantly associated with asthma (OR 2.65 (2.01–3.48)/OR 2.20 (1.53–3.16)); watching TV for 3 h but less than 5 h was a significant factor (OR 1.52 (0.93–2.46)), spending 5 h or more on social games was significantly associated with asthma (OR 0.63 (0.43–0.94)) and being a twin was a significant factor for asthma (OR 2.14 (1.52–3.03)). Additionally, the frequency of trucks passing in the street was a borderline significant factor in asthma (OR 1.66 (0.99–2.00)), having a cat at home was significant for asthma (OR 1.20 (0.86–1.66)) and exposure to tobacco smoking or smoking a water pipe were significant for asthma (OR 2.35 (1.33–4.18)/OR 1.65 (1.16–2.36)). Lastly, using paraffin as a fuel for heating was borderline significantly associated with asthma (OR 0.66 (0.36–1.20)).

Several variables in the adjusted model were significantly associated with having had ‘wheeze in the past 12 months’ in Table 5. Engaging in vigorous activity once or twice a week (OR 2.20 (1.54–3.13)) and three or more times a week (OR 1.63 (1.03–2.53)) were significantly associated with ‘wheeze in the past 12 months’. Being a twin was significantly associated with ‘wheeze in the past’ (OR 1.68 (1.03–2.78)). Trucks passing in the street frequently during the day was a significant factor for ‘wheeze in the past’ (OR 1.65 (1.01–2.69)), and having a cat at home was also a significant factor (OR 1.91 (1.34–2.73)). The use of open fires for cooking was a significant factor for ‘wheeze in the past’ (OR 2.27 (1.53–3.37)).

Several factors were associated with ‘wheeze ever’ in Table 6. In the binary regression analysis, using gas as a fuel for cooking (OR 0.72 (0.53–0.99)) and using open fires as fuel for heating the home (OR 0.53 (0.35–0.80)) were significantly associated with ‘wheeze ever’. Smoking a water pipe was significant for ‘wheeze ever’. Engaging in vigorous physical activity once or twice a week (OR 3.37 (2.69–4.22)) and three or more times a week (OR 3.48 (2.54–4.76)) were significantly associated with ‘wheeze ever’.

## 4. Discussion

This study investigated the prevalence of childhood asthma, respiratory symptoms and the associated air pollution sources in Vhembe district, South Africa. The prevalence of ‘asthma ever’ was 18.91%, while the prevalences of ‘wheeze ever’ and ‘wheeze in the past 12 months’ were 38% and 67.40%, respectively. A review in the worldwide trends in asthma and wheezing showed that although there are differences, there is a steady increase in the prevalence of asthma and wheezing in the United Kingdom (UK), Oceania and Latin American countries [10,24]. Studies have shown that wheezing is the most prominent respiratory symptom associated with asthma, which is why this study focuses on wheezing as a respiratory symptom [13].

According to the literature, the worldwide incidence of asthma among children and adolescents has risen [20]. Just as in the current study, another study has reported a high prevalence rate of asthma as estimated to be 13.14% [13,25]. The worldwide prevalence of ‘wheeze ever’ as deemed by a study conducted by Asher et al. in adolescents was 11.5%. This prevalence differed from the 7% and 21% reported in the Indian subcontinent and Oceania, respectively [10,12]. However, it is worth noting that there is an increasing prevalence and difference in risk factors for asthma in children and adolescents in low-income and middle-income countries (LMICs) compared with high-income countries [7]. The prevalence rates in this study are also consistent with those in a study conducted in the United States [26].

The prevalence of ‘asthma ever’ and ‘wheeze ever’/‘wheeze in the past’ in this study was high when compared to other ISAAC studies conducted in South Africa. A recent study conducted in preschool children in Mpumalanga showed a prevalence of ‘asthma ever’ as 2.34% and the prevalence of ‘wheeze’ as 15.14% [12]. In another study conducted in Gauteng and Mpumalanga, the prevalence of asthma and wheeze were 16.37% and 38.25%, respectively [3]. To the best of our knowledge, no study has been conducted in Vhembe district to investigate the prevalence of asthma and respiratory symptoms, especially using the ISSAC questionnaire. This may contribute to an underdiagnosis of asthma, which is a significant issue, necessitating ongoing worldwide surveillance for asthma and respiratory illnesses in order to inform health policy, management and disease prevention.

In the current study, smoking tobacco less than daily was significantly associated with ‘asthma ever’, ‘wheeze in the past 12 months’ and ‘wheeze ever’, with crude odds ratios of OR 4.07 (2.97–5.56), OR 2.40 (1.41–4.07) and OR 3.07 (2.21–4.24). These findings are in agreement with other ISAAC studies previously conducted. A study conducted in Gauteng, South Africa, showed a positive association between ‘wheeze ever’, ‘current wheeze’ and exposure to tobacco smoke [26,27]. Furthermore, a study conducted in Polokwane showed that ETS contributes to air pollution, and individuals exposed are at an increased risk of asthma [23]. Moreover, Xing and colleagues reported that the risk of asthma and wheeze are increased when children are exposed to tobacco smoke [28]. There is overwhelming evidence that links tobacco smoking with the development of asthma and respiratory symptoms, agreeing with the findings of this study. This evidence includes studies conducted by Booalayan et al., Kirenga et al. and Li et al. that also showed that adolescents who smoked were most likely to develop asthma [29,30,31]. Likewise, smoking a water pipe was associated with ‘asthma ever’ (OR 1.65 (1.16–2.36)) and ‘wheeze ever’ (OR 2.47 (1.78–3.88)). Studies conducted have also shown that smoking water pipes and electronic cigarettes was significantly associated with asthma and wheeze, although there is a perception that these products are less harmful if at all [32,33,34].

Jarvis et al. reported that the decrease in symptoms of asthma was more attributed to smoking cessation rather than the improved treatment of asthma [35]. Therefore, current legislation and regulations on tobacco smoking should be strengthened to reduce tobacco smoking and exposure, especially in public spaces [27].

In the current study, the use of paraffin as a fuel for heating and cooking was a significant risk factor for asthma. Additionally, using open fires for cooking and heating was associated with ‘asthma ever’ and wheeze. Even though there has been a significant increase in access to electricity in South Africa, most households in Limpopo still rely on alternative energy sources such as paraffin, gas, coal and wood for cooking and heating homes [23]. Socio-economic factors, along with environmental factors, play an important role in vulnerability to exposure to air pollution, as adolescents residing in areas where fires and coal are used for cooking and heating are more at risk of asthma and wheeze [23]. A study conducted by Gonzalez-Garcia et al. in Colombia showed that exposure to smoke from burning wood when cooking and fumes from gasses were risk factors significantly associated with asthma and wheeze in selected study participants [36].

In the current study, trucks passing through the street frequently during the day was a borderline significant risk factor of developing asthma. In addition, the current study also demonstrated a significant relationship between trucks passing by in the street frequently and wheezing in the past. These findings were expected, as the current study was conducted in a rural setting where majority of the air pollution sources are linked to traffic-related air pollution, dust, smoke and chemical exposures.

Furthermore, these findings are consistent with findings from other previously conducted studies. An ISAAC study conducted in over 20 centers showed a positive association between asthma and trucks passing in the streets [37]. Moreover, a study by Olaniyan and colleagues suggested that air pollution, particularly traffic-related, contributes to the development of asthma [38]. In another ISAAC study conducted in Gauteng province, trucks passing in the streets near homes of study participants posed a significant risk factor for ‘wheeze ever’, showing strong links to current and severe wheeze [27]. Additional research has shown that air pollutants, like those from car engines or exhausts, are significant causes of air pollution and have been linked to asthma and respiratory symptoms [38,39,40]. Moreover, the various ambient air quality standards for the four criterion pollutants have also been described, showed that, when it came to particulate matter, sulfur dioxide and ozone (traffic-related pollutants), South African regulations were not as strict as those set by the World Health Organization (WHO) [38]. It is recommended that policy makers focus on revising the South African air quality standards, making them more stringent.

### Strengths and Limitations

The main strength of the study was the use of a previously validated ISAAC questionnaire, which has been used globally for asthma and respiratory symptom research. The study also had a high participation rate, which further eliminated the risk of selection bias. The study limitations were that the findings were based on self-reported answers from the participants, which may lead to a misclassification of the disease, which often results in spurious associations. Furthermore, the nature of a cross-sectional study design does not provide any evidence of causality.

## 5. Conclusions

This study revealed a significant prevalence rate of asthma and wheezing among adolescents in selected schools in Vhembe district in South Africa. Moreover, we found that exposure to tobacco smoke and smoking a water pipe were associated with asthma in adolescents. Furthermore, the kind of fuel used for cooking and the frequency of trucks passing by was associated with wheezing in children and adolescents. In conclusion, our findings highlighted the significant air pollution risk factors for asthma and wheezing in children and adolescents of the Vhembe District. Hence, it is crucial to implement public health interventions that are designed to reduce air pollution exposure in order to alleviate asthma and wheezing in adolescents. In addition, the outcome of our investigation also highlights the urgent need for effective public health interventions aimed at bringing about a better understanding of asthma, asthma management and lastly air pollution and its effect on respiratory health. The results of this study will contribute to the growing body of knowledge that suggests air pollution plays a critical role in respiratory health. Future research may investigate other air pollution sources, i.e., the impact of vehicle types on asthma and respiratory symptoms.

## Figures and Tables

**Figure 1 ijerph-21-01536-f001:**
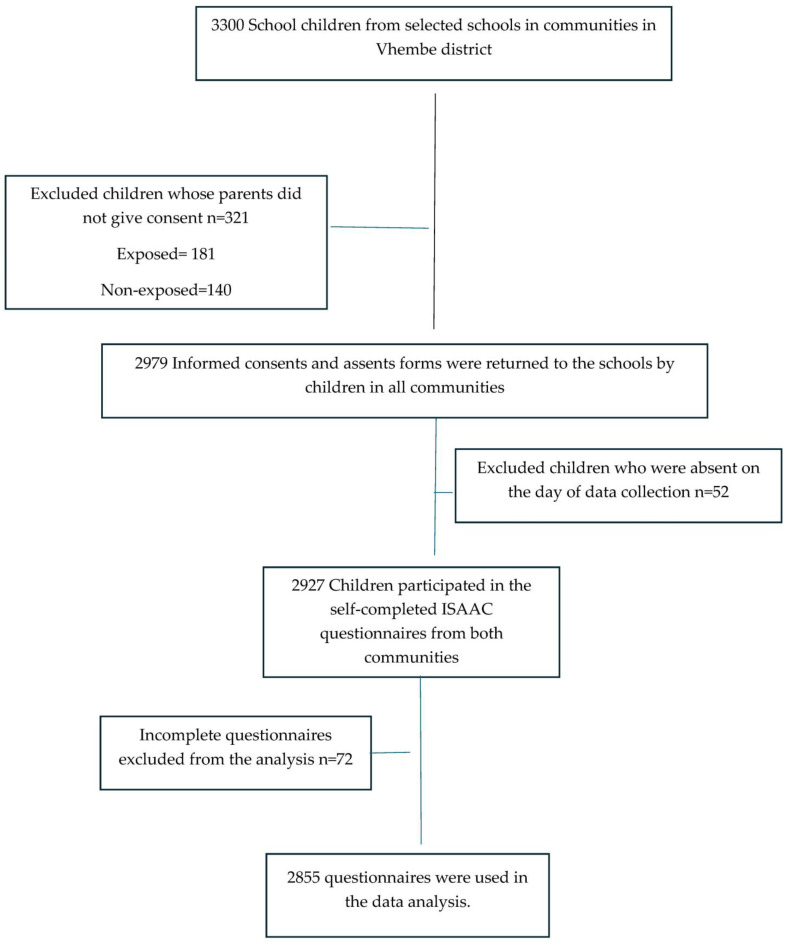
Flow chart of procedures followed and the participation rate.

**Table 1 ijerph-21-01536-t001:** The demographic characteristics, health outcomes and risk factors of study participants (*n* = 2855).

Variable	*N*	Percentage (%)
**1. Gender**		
Female	1632	57.16 (57.89)
Male	1187	41.58 (42.11)
Missing	36	1.26
**2. Age**		
13 years	1965	68.83
14 years	857	30.02
Missing	33	1.16
**3. Community**		
Exposed	1751	61.33 (61.46)
Non-exposed	1098	38.46 (38.54)
Missing	6	0.21
**4. Born in study area**		
Yes	2052	71.87 (76.06)
No	646	22.63 (23.94)
Missing	157	5.50
**5. School**		
Primary	1295	45.36
Secondary	1560	54.64

**Table 2 ijerph-21-01536-t002:** The prevalence of ‘asthma ever’, ‘wheeze ever’ and ‘wheeze in the past 12 months’ in the study participants (*n* = 2855).

Variable	*N*	Percentage (%)
**Asthma Ever**		
Yes	533	18.67 (18.91)
No	2286	80.07 (81.09)
Missing	36	1.26
**Wheeze Ever**		
Yes	1076	37.69 (38.06)
No	1751	61.33 (61.94)
Missing	28	0.98
**Wheeze in the Past 12 Months**		
Yes	705	24.69 (67.40)
No	341	11.94 (32.50)
Missing	18.09	63.36

**Table 3 ijerph-21-01536-t003:** Risk factors of the study participants (*n* = 2855).

Variable	*N*	Percentage (%)
**Fuel used for cooking in the house**		
Electricity	1628	57.02 (58.18)
Gas	393	13.77 (14.05)
Paraffin	113	3.96 (4.04)
Open fire (wood, coal)	664	23.26 (23.73)
Missing	57	2.00
**Fuel used for heating in the house**		
Electricity	1848	64.73 (66.47)
Gas	331	11.59 (11.91)
Paraffin	411	14.40 (14.78)
Open fires (wood, coal)	190	6.65 (6.83)
Missing	75	2.63
**Frequency of trucks passing in the street**		
Never	1101	38.56 (39.58)
Seldom	637	22.31 (22.90)
Frequently through the day	508	17.79 (18.26)
Almost the whole day	536	18.77 (19.27)
Missing	73	2.56
**Currently smoking tobacco**		
Not at all	2433	85.22 (88.63)
Less than daily	166	5.81 (6.62)
Daily	142	4.97 (4.75)
Missing	114	3.99
**Smoke water pipe**		
Yes	290	10.16 (10.50)
No	2471	86.55 (89.50)
Missing	94	3.29
**Have a dog at home**		
Yes	1475	51.66 (53.75)
No	1269	44.45 (46.25)
Missing	111	3.89
**Have a cat at home**		
Yes	756	26.48 (27.28)
No	2015	70.58 (72.72)
Missing	84	
**Are you a twin**		
Yes	280	9.81 (10.06)
No	2503	87.67 (89.94)
Missing	72	2.52
**Use paracetamol**		
Never	1712	59.96 (62.90)
At least once a year	653	22.87 (24.03)
At least once a month	353	12.36 (12.99)
Missing	137	4.80
**Physical activity**		
Never or only occasionally	1668	58.42 (62.90)
Once or twice per week	704	24.66 (26.55)
Three or more times a week	280	9.81 (10.56)
Missing	203	7.11
**Games, social**		
Less than 1 h	1133	39.68 (41.23)
1 h but less than 3 h	729	25.53 (26.53)
3 h but less than 5 h	498	17.44 (18.12)
5 h or more	388	13.59 (14.12)
Missing	107	3.75
**Watching television**		
Less than 1 h	1165	40.81 (42.94)
1 h but less than 3 h	903	31.63 (33.28)
3 h but less than 5 h	304	10.65 (11.21)
5 h or more	341	11.94 (12.57)
Missing	142	4.97

**Table 4 ijerph-21-01536-t004:** The prevalence of ‘asthma ever’ along with crude and adjusted odds ratios.

Variable	Total ^a^	Asthma Ever (%)	Crude OR		Adjusted OR ^b^	
			(95% CI)	*p*	(95% CI)	*p*
**How many times a week do you engage in vigorous physical activity?**						
Never or only occasionally	187	11.35%	1		1	
Once or twice per week	231	32.95%	3.83 (3.08–4.77)	**0.000**	2.65 (2.01–3.48)	**0.000**
Three or more times a week	80	28.78%	3.15 (2.33–4.26)	**0.000**	2.20 (1.53–3.16)	**0.000**
**During a normal week of 7 days, how many hours a day (24 h) do you watch television?**						
Less than 1 h	202	17.50%	1		1	
1 h but less than 3 h	199	22.28%	1.35 (1.08–1.68)	**0.007**	1.38 (0.97–1.97)	0.118
3 h but less than 5 h	57	19%	1.10 (0.79–1.53)	0.547	1.52 (0.93–2.46)	**0.021**
5 h or more	55	16.62%	0.93 (0.67–1.30)	0.707	0.96 (0.58–1.60)	0.708
**During a normal week of 7 days, how many hours a day (24 h) do you spend on any social games?**						
Less than 1 h	184	16.53%	1		1	
1 h but less than 3 h	127	17.52%	1.07 (0.83–1.37)	0.582	0.70 (0.50–0.95)	**0.023**
3 h but less than 5 h	127	25.81%	1.75 (1.35–2.27)	**0.000**	0.81 (0.53–1.23)	0.113
5 h or more	79	20.63%	1.31 (0.97–1.76)	0.070	0.63 (0.43–0.94)	**0.023**
**Are you a twin?**						
No	394	15.94%	1		1	
Yes	114	40.71%	3.62 (2.78–4.70)	**0.000**	2.14 (1.52–3.03)	**0.000**
**How often do trucks pass through the street where you live on weekdays?**						
Never	149	13.71%	1		1	
Seldom	119	18.86%	1.46 (1.12–1.90)	**0.005**	1.13 (0.74–1.73)	0.929
Frequently through the day	143	28.60%	2.52 (1.94–3.26)	**0.000**	1.66 (0.99–2.00)	**0.055**
Almost the whole day	110	20.79%	1.65 (1.25–2.16)	**0.000**	1.64 (0.99–1.94)	**0.056**
**In the past 12 months how often, on average, have you taken paracetamol?**						
Never	225	13.31%	1		1	
At least once a year	193	29.56%	2.73 (2.19–3.40)	**0.000**	1.01 (0.70–1.46)	0.096
At least once a month	95	27.78%	2.50 (1.90–3.29)	**0.000**	1.79 (1.28–2.51)	**0.001**
**In the past 12 months, have you had a cat in your home?**						
No	214	28.53%	1		1	
Yes	303	15.24%	2.22 (0.81–2.71)	**0.000**	1.20 (0.86–1.66)	**0.013**
**In the past 12 months, have you had a dog in your home?**						
No	311	21.21%	1		1	
Yes	196	15.74%	1.44 (1.18–1.75)	**0.000**	1.08 (0.84–1.37)	0.522
**In the past, have you smoked tobacco on a daily basis, less than daily, or not at all?**						
Not at all	389	16.18%	1		1	
Less than daily	73	44.51%	4.15 (2.99–5.75)	**0.000**	1.84 (1.08–3.16)	**0.025**
Daily	54	38.03%	3.17 (2.22–4.53)	**0.000**	2.35 (1.33–4.18)	**0.003**
**Do you currently smoke tobacco?**						
Not at all	390	16.29%	1		1	
Less than daily	80	44.20%	4.07 (2.97–5.56)	**0.000**	2.21 (1.10–4.41)	0.124
Daily	40	30.77%	2.28 (1.54–3.36)	**0.000**	1.85 (0.75–4.53)	0.850
**Do you smoke a water pipe?**						
No	101	35.31%	1		1	
yes	420	17.21%	2.62 (2.01–3.42)	**0.000**	1.65 (1.16–2.36)	**0.005**
**What fuel is usually used for cooking?**						
Electricity	254	15.84%	1		1	
Gas	104	26.74%	1.93 (1.49–2.51)	**0.000**	1.13 (0.72–1.79)	0.575
Paraffin	47	42.34%	3.90 (2.61–5.82)	**0.000**	1.70 (0.97–2.88)	**0.059**
Open fires	121	18.33%	1.19 (0.94–1.51)	0.147	1.33 (0.92–1.92)	0.115
**What fuel is usually used for heating?**						
Electricity	323	17.72%	1		1	
Gas	101	31.08%	2.09 (1.60–2.72)	**0.000**	0.79 (0.49–1.27)	0.764
Paraffin	69	16.83%	0.93 (0.70–1.24)	0.669	0.48 (0.29–1.00)	**0.051**
Open fires	29	15.43%	0.84 (0.56–1.28)	0.431	0.66 (0.36–1.20)	0.225

^a^ Totals for each risk factor are different due to differences in missing values. Only variables significant in univariate were included in the adjusted model. ^b^ Model adjusted for all the variables in this table. Values that are less the 0.05 are statistically significant and are in bold.

**Table 5 ijerph-21-01536-t005:** The prevalence of self-reported ‘wheeze in the past 12 months’ along with crude and adjusted odds ratios.

Variable	Total ^a^	Wheeze In the Past (%)	Crude OR		Adjusted OR ^b^	
			(95% CI)	*p*	(95% CI)	*p*
**How many times a week do you engage in vigorous physical activity?**						
Never or only occasionally	230	55.69%	1		1	
Once or twice per week	312	77.41%	2.72 (2.01–3.69)	**0.000**	2.20 (1.54–3.13)	**0.000**
Three or more times a week	105	67.74%	1.67 (1.13–2.46)	**0.010**	1.63 (1.03–2.53)	**0.033**
**During a normal week of 7 days, how many hours a day (24 h) do you watch television?**						
Less than 1 h	270	63.98%	1		1	
1 h but less than 3 h	252	69.23%	1.26 (0.93–1.70)	0.121	1.07 (0.67–1.71)	0.710
3 h but less than 5 h	79	74.52%	1.64 (1.01–2.66)	**0.042**	1.44 (0.71–2.92)	0.264
5 h or more	85	70.83%	1.36 (0.87–2.12)	0.164	1.08 (0.77–1.87)	0.772
**During a normal week of 7 days, how many hours a day (24 h) do you spend on any social games?**						
Less than 1 h	240	63.49%	1		1	
1 h but less than 3 h	180	65.21%	1.07 (0.77–1.49)	0.649	0.94 (0.63–1.41)	0.799
3 h but less than 5 h	162	76.77%	1.90 (1.29–2.78)	**0.001**	0.97 (0.59–1.58)	0.915
5 h or more	107	68.15%	1.36 (0.82–1.82)	0.304	0.94 (0.57–1.53)	0.816
**Are you a twin?**						
No	563	65.69%	1		1	
Yes	121	77.07%	1.75 (1.17–2.61)	**0.006**	1.68 (1.03–2.78)	**0.037**
**How often do trucks pass through the street where you live on weekdays?**						
Never	207	60.70%	1		1	
Seldom	154	66.95%	1.31 (0.92–1.86)	0.129	1.43 (0.93–2.20)	0.102
Frequently through the day	183	70.77%	2.19 (1.51–3.18)	**0.000**	1.65 (1.01–2.69)	**0.045**
Almost the whole day	145	66.51%	1.28 (0.90–1.83)	0.166	1.20 (0.78–1.83)	0.397
**In the past 12 months how often, on average, have you taken paracetamol?**						
Never	300	59.88%	1		1	
At least once a year	241	71.72%	1.69 (1.26–2.28)	**0.000**	1.13 (0.78–1.62)	0.510
At least once a month	125	78.12%	2.29 (1.57–3.62)	**0.000**	1.98 (1.20–3.26)	**0.007**
**In the past 12 months, have you had a cat in your home?**						
No	398	60.94%	1		1	
Yes	278	78.08%	0.43 (1.69–3.07)	**0.000**	1.91 (1.34–2.73)	**0.000**
**In the past, have you smoked tobacco on a daily basis, less than daily, or not at all?**						
Not at all	546	66.02%	1		1	
Less than daily	84	82.35%	2.40 (1.41–4.07)	**0.001**	1.55 (0.81–2.96)	0.184
Daily	43	55.84%	0.65 (0.40–1.04)	0.075	0.55 (0.29–1.03)	0.065
**What fuel is usually used for cooking?**						
Electricity	351	62.45%	1	1		
Gas	117	68.02%	1.27 (0.88–1.83)	0.184	0.84 (0.53–1.35)	0.497
Paraffin	48	82.75%	2.88 (1.42–5.82)	**0.003**	1.94 (0.78–4.80)	0.152
Open fires	177	74.05%	1.71 (1.22–2.40)	**0.002**	2.27 (1.53–3.37)	**0.000**

^a^ Totals for each risk factor are different due to differences in missing values. Only variables significant in univariate were included in the adjusted model. ^b^ Model adjusted for all the variables in this table. Values that are less the 0.05 are statistically significant and are in bold.

**Table 6 ijerph-21-01536-t006:** The prevalence of self-reported ‘wheeze ever’, along with crude and adjusted odds ratios.

Variable	Total ^a^	Wheeze Ever ^a^ (%)	Crude OR		Adjusted OR ^b^	
			(95% CI)	*p*	(95% CI)	*p*
**Fuel used for cooking in the house**						
Electricity	581	36.11%	1		1	
Gas	176	45.01%	1.44 (1.15–1.81)	**0.001**	0.72 (0.53–0.99)	**0.045**
Paraffin	60	53.10%	2.00 (1.36–2.93)	**0.000**	0.78 (0.47–1.29)	0.338
Open fires (wood, coal)	243	36.82%	1.03 (0.85–1.24)	0.750	1.70 (0.84–1.37)	0.546
**Fuel used for heating in the house**						
Electricity	668	36.46%	1		1	
Gas	171	51.82%	1.87 (1.48–2.37)	**0.000**	1.12 (0.81–1.55)	0.481
Paraffin	160	39.60%	1.14 (0.92–1.43)	0.237	1.17 (0.88–1.56)	0.275
Open fires (wood, coal)	54	28.42%	0.69 (0.49–0.96)	**0.028**	0.53 (0.35–0.80)	**0.003**
**Frequency of trucks passing in the street**						
Never	356	32.60%	1		1	
Seldom (not often)	235	37.24%	1.22 (0.99–1.50)	**0.051**	0.87 (0.67–1.14)	0.326
Frequently through the day	241	48.01%	1.90 (1.53–2.37)	**0.000**	1.06 (0.79–1.43)	0.677
Almost the whole day	222	41.57%	1.14 (0.87–1.49)	**0.000**	0.91 (0.65–1.27)	0.333
**Smoking tobacco**						
Not at all	852	35.35%	1		1	
Less than daily	104	62.65%	3.07 (2.21–4.24)	**0.000**	0.89 (0.52–1.50)	0.672
Daily	77	54.23%	2.17 (1.54–3.04)	**0.000**	0.96 (0.55–1.67)	0.891
**Smoking a water pipe**						
No	860	35.13%	1		1	
Yes	179	61.94%	3.00 (2.33–3.86)	**0.000**	2.47 (1.78–3.44)	**0.000**
**Have a dog at home**						
No	413	32.80%	1		1	
Yes	605	41.52%	1.84 (1.55–2.18)	**0.000**	1.04 (0.85–1.26)	0.678
**Have a cat at home**						
No	671	33.75%	1		1	
Yes	364	48.34%	1.84 (1.55–2.18)	**0.000**	0.97 (0.78–1.21)	0.826
**Use paracetamol**						
Never	517	30.38%	1		1	
At least once a year	345	53.41%	2.63 (2.18–3.16)	**0.000**	1.79 (1.41–2.28)	**0.000**
At least once a month	162	46.69%	2.00 (1.58–2.54)	**0.000**	1.72 (1.29–2.29)	**0.000**
**Physical activity**						
Never or only occasionally	421	25.44%	1		1	
Once or twice per week	415	59.54%	4.31 (3.57–5.20)	**0.000**	3.37 (2.69–4.22)	**0.000**
Three or more times a week	162	58.48%	4.12 (3.17–5.37)	**0.000**	3.48 (2.54–4.76)	**0.000**
**Games, social**						
Less than 1 h	396	35.14%	1		1	
1 h but less than 3 h	280	38.94%	1.17 (0.97–1.43)	0.098	1.19 (0.88–1.62)	0.240
3 h but less than 5 h	216	43.81%	1.43 (1.15–1.78)	**0.001**	1.11 (0.77–1.59)	0.560
5 h or more	160	41.56%	1.31 (1.03–1.66)	**0.024**	1.34 (0.92–1.95)	0.120
**Watching television**						
Less than 1 h	433	37.46%	1		1	
1 h but less than 3 h	377	42.12%	1.21 (1.01–1.45)	**0.032**	1.21 (0.90–1.61)	0.281
3 h but less than 5 h	106	35.10%	0.90 (0.69–1.17)	0.450	1.22 (0.82–1.83)	0.415
5 h or more	125	37.31%	0.99 (0.77–1.28)	0.962	1.12 (0.76–1.66)	0.631

^a^ Totals for each risk factor are different due to differences in missing values. ^b^ Model adjusted for all the variables in this table. Values that are less the 0.05 are statistically significant and are in bold.

## Data Availability

The ethical approval we received limits us from sharing the data publicly. Raw data analyzed are available upon reasonable request to the authors.

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
