# Peer review of "The Prevalence of Childhood Asthma, Respiratory Symptoms and Associated Air Pollution Sources Among Adolescent Learners in Selected Schools in Vhembe District, South Africa"

_ijerph, 2024, doi:10.3390/ijerph21111536_

Round 1

Reviewer 1 Report

Comments and Suggestions for Authors

The article highlights factors contributing to wheezing and asthma in adolescents in the Vhembe district, an area subjected to high pollution levels. The study holds significant importance. I agree with the author, but we should consider making modifications and additions to the relevant content before publication. 

1. Table 2 (line 160) currently presents only the total count of each item. It is recommended to add two additional columns of data for the exposed and non-exposed groups and conduct statistical analysis. 

2. It is advisable to include a detailed description of the statistical methods related to Table 2 in the 2.6. Data management and statistical analysis section (lines 133-148) of the Methods.

Reviewer 2 Report

Comments and Suggestions for Authors

Manuscript ID- ijerph-3287404

The authors of this manuscript (MS) by Funzani Rathogwa-Takalani et al., this study investigated the prevalence of childhood asthma, respiratory symptoms with their associated air pollution sources among adolescents aged years residing in a Malar-ia-endemic region. A cross-sectional survey was conducted with 2855 adolescents from fourteen (14) selected schools in communities exposed to high levels of air pollution from Indoor residual spraying (IRS). A statistical software STATA 17 was used to analyse the data. Multivariate logistic regression was used to determine the relationship between air pollution sources and childhood asthma/symptoms. The prevalence of asthma, wheeze ever and wheeze in the past were 18.91%, 37.69% and 24.69%, respectively. The results from the adjusted multivariate logistic regression model indicated that exposure to tobacco smoke (OR=1.84; 95% CI: 1.08-3.16), smoking a water pipe (OR= 1.65; 95% CI: 1.16 - 2.36), the use of paraffin as fuel for heating (OR= 1.70; 95% CI: 0.97- 2.88) and cooking (OR= 0.48; 95% CI: 0.29 -1.00) were significant risk factors for asthma. School children living in these communities had an increased risk of developing asthma and presenting with wheezing due to exposure environmental air pollution sources.

The abstract, methodology, discussion, and overall structure are acceptable based on this MS, but there is less explanation in detail. I think before publication, some aspect of the strategy can be revised to improve the quality of the MS.  The authors, in my opinion, should expound on their findings in the contents. The manuscript may be published after major changes. The following recommendations should be addressed in the revised MS.

Comments and Suggestions

1.       Introduction and line 153: I think the author should explain more about Indoor Residual Spraying (IRS) for malaria control. How does it relate to air pollution?

2.       Line 85-87: The author wrote study among 13 to 14 years school children but why you choose seven primary schools

3.       Figure 1: Please change fronts in the flow chart

4.       Line 119: Why 12 months?

5.       Table 1: variables, it should be 1 to 5. Please check 5. Born in study area or 4. Born in study area?

6.       Are the majority of study participants females in secondary school?

7.       Line 123: How was the frequency of cooking? What about children who help their parents cook and do not? Did you ask or consider?

8.       Line 125: How many pet in each home? Did you ask or consider? Please explain more details.

9.       Result: Table 1: What type of gas? 

10.    Are the health outcomes and risk factors difference of the study participants in primary and secondary school?

11.    Table 3: I wonder about the question about smoke tobacco, the author inquired about a family member or children.

12.    Table 4: The author asked “How often do trucks pass through the street where you live on “weekdays? Why does the author consider only trucks? What about other vehicles?

13.    Discussion: The author should elaborate more about the discussion, such as trying to compare your study with other references. How do pollutants from tobacco smoke and water pipe smoking contribute to the development of asthma in adolescents?

14.    Discussion: The author discusses indoor residual spraying (IRS) for malaria control in the abstract and results (line 153), but it appears you have not included any information in the discussion.

15.    Conclusion: The conclusions could be further developed, there is a lot of interesting data in the article.

16.    All references mentioned in the reference list are cited in the texts, and vice versa. Author can put more references.

Reviewer 3 Report

Comments and Suggestions for Authors

I would like to take this opportunity to express my sincere gratitude for the opportunity to participate in this review. It is a cross-sectional study conducted on children to analyze the prevalence of asthma and respiratory symptoms, such as wheezing, in children exposed to sources of environmental air pollution, including various fuels used for cooking or heating and tobacco smoke, among others. It is an exciting and well-written article that could be a strong candidate for publication in IJERPH. However, the authors have to address some aspects to ensure the study is comprehensive and accurate.

1.    The authors discuss asthma and wheezing from lines 45 to 51 in the paragraph. A brief description of the relationship between the two would enhance understanding and provide better context. This addition could greatly enrich the readers' comprehension of the complexities involved in respiratory conditions in children.

2.    The authors discuss ISAAC throughout the text. However, whether this is a questionnaire or a cohort study is unclear. This doubt is repeated throughout the text, so I suggest that this aspect be revised and that more detail be given as to what it consists of. Additionally, the text mentions that the validated English version of ISAAC has been used in this regard. However, I have a question: Was ISAAC designed initially as an English questionnaire that has been validated, or is it a translation of a questionnaire validated in another language? If it is the latter, I would appreciate clarification on whether the translation has been validated. This information would significantly enhance our understanding of the study's methodology.

3.    The final paragraph in the statistical analysis section states that univariate and multivariate logistic regression were conducted. However, the results presented suggest that the analysis aligns more closely with binary logistic regression. I encourage the authors to revisit this aspect and offer a more precise description of the regression model employed. Your role in this aspect is crucial for the final approval of the manuscript.

4.    In Table 1, in the community section, it is not clear what is meant by 'exposed' and 'non-exposed,' particularly since the study examines various sources of exposure. Please clarify this aspect in the text.

5.    I would suggest that the authors consider splitting Table 2 into two separate tables, one for health outcomes and another for risk factors, to enhance clarity. I would also recommend reviewing the values for the number of missing entries under “wheezing in the past 12 months.” Is there any specific reason that might explain why this number is so high?

6.    In lines 164-166, there appears to be a discrepancy regarding the value for paraffin cooking, as it does not align with what is presented in Table 2. I kindly request that the authors review this to ensure consistency. Additionally, it would be beneficial to address other risk factors, particularly the impact of tobacco smoke, as this is an especially relevant aspect of the discussion. 

7.    Section 3.2 presents a collection of results, but the type of regression employed remains unclear. As noted in point 3 of this review, I urge the authors to revisit this whole section and rewrite it for greater clarity (refer to lines 215-216, for instance).

8.    In line 261 the authors introduce the term ETS. What do they mean by it?

9.    I recommend changing the subheading "Limitations" to "Strengths and Limitations." This adjustment would offer a more balanced viewpoint and emphasize both the study's positive aspects and constraints, thereby improving the manuscript's overall clarity.

Once these changes have been made, I am convinced the article is suitable for publication.

Round 2

Reviewer 2 Report

Comments and Suggestions for Authors

Second round:

The manuscript has been revised from scientific points of view quite thoroughly. On the other hand, there are still typos and technical errors in the MS, which should be corrected before publication. Please see detailed comments below.

Specific comments- (abbreviation, L-line)

- L18: Please, provide “version” before 17.

- L31: Please change “respiratory symptoms” to Respiratory symptoms.

- L72: please change “District” to district.

- L143: Please, also add here “REDCap” after the full term.

- Table 1: “N” instead “N

- Table 1, 2, 3: Variables or Variable, Please, correct the English.

- L276: Please change “Tobacco smoke” to tobacco smoke.

- L285: electronic cigarette is better than e-cigarette.

- L319: The sentence starting with “The former study also described the various ambient air…” is not clear, and it should be clarified for the reader.
